# Improved Variational Bayesian Phylogenetic Inference with Normalizing Flows

**Cheng Zhang**

School of Mathematical Sciences and Center for Statistical Science
Peking University, Beijing, China
chengzhang@math.pku.edu.cn

## Abstract

Variational Bayesian phylogenetic inference (VBPI) provides a promising general variational framework for efficient estimation of phylogenetic posteriors. However, the current diagonal Lognormal branch length approximation would significantly restrict the quality of the approximating distributions. In this paper, we propose a new type of VBPI, VBPI-NF, as a first step to empower phylogenetic posterior estimation with deep learning techniques. By handling the non-Euclidean branch length space of phylogenetic models with carefully designed permutation equivariant transformations, VBPI-NF uses normalizing flows to provide a rich family of flexible branch length distributions that generalize across different tree topologies. We show that VBPI-NF significantly improves upon the vanilla VBPI on a benchmark of challenging real data Bayesian phylogenetic inference problems. Further investigation also reveals that the structured parameterization in those permutation equivariant transformations can provide additional amortization benefit.

## 1  Introduction

As a powerful statistical tool that has revolutionized modern molecular evolutionary analysis, Bayesian phylogenetic inference has been widely used for tasks ranging from genomic epidemiology (Neher and Bedford, 2015; Sun et al., 2020) to conservation genetics (DeSalle and Amato, 2004). Given properly aligned sequence data (e.g., DNA, RNA or protein sequences) and a model of evolution, Bayesian phylogenetics provides principled approaches to quantify the uncertainty of the evolutionary process in terms of the posterior probabilities of phylogenetic trees (Huelsenbeck et al., 2001). A commonly used Bayesian phylogenetic inference method is random-walk Markov chain Monte Carlo (MCMC), which was introduced to the community in the late 1990's (Yang and Rannala, 1997; Mau et al., 1999; Huelsenbeck and Ronquist, 2001). However, random-walk MCMC has been fundamentally limited as it often exhibits low exploration efficiency and requires long runs to deliver accurate posterior estimates due to the complexity of tree space. Although many advanced methods for posterior sampling have been proposed recently, including Hamiltonian Monte Carlo (Duane et al., 1987; Neal, 2011), it is not straightforward to extend these methods to phylogenetic models due to the composite structure of tree space, i.e., a combination of discrete variables (e.g., tree topologies) and continuous variables (e.g., branch lengths) (Dinh et al., 2017b).

Variational inference (VI) (Jordan et al., 1999; Wainwright and Jordan, 2008; Blei et al., 2017) is an alternative approximate Bayesian inference method that is growing in popularity. Unlike MCMC methods, VI seeks the best approximation to the true posterior from a family of tractable distributions. By transforming the inference problem into an optimization problem, VI tends to be faster and easier to scale to large data (Blei et al., 2017). In recent years, many efforts have been made to harness VI for phylogenetic inference (Zhang and Matsen IV, 2019; Dang and Kishino, 2019; Fourment and Darling, 2019), among which *variational Bayesian phylogenetic inference* (VBPI) proposed by

Zhang and Matsen IV (2019) provides a promising general framework that allows joint learning of phylogenetic trees with branch lengths. At the core of VBPI lie *subsplit Bayesian networks* (SBNs) (Zhang and Matsen IV, 2018), an expressive probabilistic graphical model for distributions over the tree topology space, and a structured amortization of the branch lengths over different tree topologies. With guided exploration in the tree space (enabled by SBNs) and joint learning of the branch length distributions across tree topologies (via amortization), VBPI provides competitive performance to MCMC with much less computation. However, the diagonal Lognormal branch length distribution currently used in VBPI might not be flexible enough to resemble the true posterior distributions.

A powerful framework for building flexible approximating distributions is normalizing flows (NFs) (Rezende and Mohamed, 2015; Dinh et al., 2017a; Kingma et al., 2016; Papamakarios et al., 2019). Starting from a simple base distribution with a tractable probability density function, NFs apply a sequence of invertible transformations, often parameterized by neural networks, to obtain a more flexible distribution. These flow-based approximating distributions enjoy many advantages such as efficient sampling, exact likelihood evaluation, and low-variance Monte Carlo gradient estimates when the base distribution is reparameterizible, making them ideal for variational inference. While efficient, current NFs are primarily designed for distributions on Euclidean space, and as a result, these approaches are ill-equipped for phylogenetic models where the tree topology and the branch lengths are intertwined in a rather complex and non-Euclidean fashion.

In this paper, we propose a new type of VBPI, VBPI-NF (Normalizing Flows), which incorporates normalizing flows for more expressive branch length approximations. More specifically, we develop permutation equivariant normalizing flows to deal with the non-Euclidean branch length space across different tree topologies, with structured parameterization based on the local topologies of trees. Inference using VBPI-NF provides tighter lower bounds and can be performed the same way as in Zhang and Matsen IV (2019). Experiments on a benchmark of challenging real data Bayesian phylogenetic inference problems demonstrate the significant improvement of VBPI-NF over the vanilla VBPI. Further investigation also shows that the transformations in these permutation equivariant normalizing flows can provide additional amortization benefit while improving approximation.

## 2   Background

**Notation**   A phylogenetic tree is denoted as $(\tau, \boldsymbol{q})$, where $\tau$ is a bifurcating tree topology that represents the evolutionary diversification of the species and $\boldsymbol{q}$ is a vector of the associated non-negative branch lengths for the edges of $\tau$. The leaf nodes of $\tau$ correspond to the labeled observed species and the internal nodes represent the unobserved characters (e.g., DNA bases) of the ancestral species. An edge incident to a leaf is called a pendant edge, and any other edge is called an internal edge. Let $\mathcal{X}$ be the set of leaf labels. A nonempty subset of $\mathcal{X}$ is called a *clade*. Let $\succ$ be a total order on clades (e.g., lexicographical order). A *subsplit* $(W, Z)$ of a clade $X$ is an ordered bipartition of $X$, that is $W \cup Z = X, W \cap Z = \emptyset$ and $W \succ Z$. The *split* $e/\tau$ of edge $e$ on a tree topology $\tau$ is the bipartition of $\mathcal{X}$ formed by the clades from different sides of $e$; The *primary subsplit pairs* (PSPs) $e/\!\!/\tau$ of

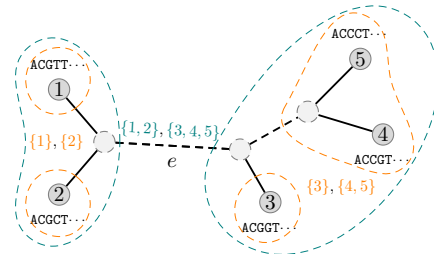

Figure 1: An unrooted phylogenetic tree and its local topological structures. 1,2,3,4,5 are the labels for the species. The *solid* edges are pendant edges and the *dashed* are internal edges. Here, the split of an internal edge $e$ is $(\{1, 2\}, \{3, 4, 5\})$ and the PSPs of edge $e$ contains $(\{1\}, \{2\})|(\{1, 2\}, \{3, 4, 5\})$ and $(\{3\}, \{4, 5\})|(\{1, 2\}, \{3, 4, 5\})$.

$e$ on a tree topology $\tau$ is a set of the conditional subsplits of the clades from different sides of $e$ given the split of $e$. See Figure 1 for an example and Zhang and Matsen IV (2018, 2019) for more details on splits and PSPs. The transition probability $P_{i \to j}(t)$ from character $i$ to character $j$ along an edge of length $t$ is often defined by a continuous-time Markov model (e.g., Jukes and Cantor (1969)). Let $E(\tau)$ be the set of edges of $\tau$, $\rho$ be the root node (or any internal node if the tree is unrooted and the Markov model is reversible), and $\eta$ be the stationary distribution of the Markov model.

**Phylogenetic posterior**   Let $\boldsymbol{Y} = \{Y_1, Y_2, \ldots, Y_M\} \in \Omega^{N \times M}$ be the observed sequences (with characters in $\Omega$) of length $M$ over $N$ species. Assuming different sites are independent and identically

distributed, the probability of observing $\boldsymbol{Y}$ given the phylogenetic tree takes the form

$$p(\boldsymbol{Y}|\tau, \boldsymbol{q}) = \prod_{i=1}^{M} \sum_{a^i} \eta(a_\rho^i) \prod_{(u,v) \in E(\tau)} P_{a_u^i \to a_v^i}(q_{uv})$$

where $a^i$ ranges over all extensions of $Y_i$ to the internal nodes with $a_u^i$ being the assigned character of node $u$. This phylogenetic likelihood function can be efficiently evaluated through the pruning algorithm (Felsenstein, 2003). Given a prior distribution $p(\tau, \boldsymbol{q})$ of the tree topology and the branch lengths, Bayesian phylogenetics then amounts to properly estimate the phylogenetic posterior

$$p(\tau, \boldsymbol{q}|\boldsymbol{Y}) = \frac{p(\boldsymbol{Y}|\tau, \boldsymbol{q})p(\tau, \boldsymbol{q})}{p(\boldsymbol{Y})} \propto p(\boldsymbol{Y}|\tau, \boldsymbol{q})p(\tau, \boldsymbol{q}).$$

**Variational Bayesian phylogenetic inference** With a support of the conditional probability tables (CPTs) (i.e., the parameters of SBNs) acquired via fast heuristic bootstrap methods (Minh et al., 2013), VBPI posits a flexible family of approximating distributions over the joint latent space of phylogenetic models using the product of an SBN-based distribution $Q_\phi(\tau)$ over the tree topologies and a diagonal Lognormal distribution $Q_\psi(\boldsymbol{q}|\tau)$ over the branch lengths. The branch length approximation $Q_\psi(\boldsymbol{q}|\tau)$ is amortized over the tree topologies via shared local structures (i.e., splits and PSPs), which are available from the support of CPTs. The best approximation is then obtained by maximizing the following multi-sample lower bound

$$L^K(\boldsymbol{\phi}, \boldsymbol{\psi}) = \mathbb{E}_{Q_{\phi,\psi}(\tau^{1:K}, \boldsymbol{q}^{1:K})} \log \left( \frac{1}{K} \sum_{i=1}^{K} \frac{p(\boldsymbol{Y}|\tau^i, \boldsymbol{q}^i)p(\tau^i, \boldsymbol{q}^i)}{Q_\phi(\tau^i)Q_\psi(\boldsymbol{q}^i|\tau^i)} \right) \leq \log p(\boldsymbol{Y}) \qquad (1)$$

where $Q_{\phi,\psi}(\tau^{1:K}, \boldsymbol{q}^{1:K}) = \prod_{i=1}^{K} Q_\phi(\tau^i)Q_\psi(\boldsymbol{q}^i|\tau^i)$. Compared to standard evidence lower bound that uses one sample, using multiple samples encourages exploration (especially in the tree topology space) and improves the tightness of the lower bound (Burda et al., 2016). See section A and B in the supplement and Zhang and Matsen IV (2018, 2019) for more details on SBNs and VBPI.

**Normalizing flows** Normalizing flow (Rezende and Mohamed, 2015; Dinh et al., 2017a; Kingma et al., 2016; Papamakarios et al., 2019) is a change of variable procedure for constructing complex and tractable distributions by transforming probability densities through a sequence of invertible mappings. Let $\boldsymbol{z}_0$ be the initial random variable that follows a simple base distribution $q_0(\boldsymbol{z}_0)$, and $\{\boldsymbol{f}_\ell\}_{\ell=1,\ldots,L}$ be the sequence of invertible parameterized transformations: $\boldsymbol{z}_\ell = \boldsymbol{f}_\ell(\boldsymbol{z}_{\ell-1})$, $\ell = 1, \ldots, L$. As long as the determinants of the Jacobians of these transformations are easily computable, we can still compute the probability density function of the last iterate:

$$q_L(\boldsymbol{z}_L) = q_0(\boldsymbol{z}_0) \prod_{\ell=1}^{L} \left| \det \frac{\partial \boldsymbol{z}_\ell}{\partial \boldsymbol{z}_{\ell-1}} \right|^{-1} \qquad (2)$$

Due to the law of the unconscious statistician (LOTUS), NFs provide a rich family of approximating distributions for VI that can be easily trained with efficient Monte Carlo gradient estimates.

## 3 Proposed method

While VBPI proves effective for Bayesian phylogenetics, the diagonal Lognormal branch length approximation and simple amortization via splits/PSPs as proposed in Zhang and Matsen IV (2019) remain too restrictive to fully capture the complexity of real data branch length posteriors. In this section, we propose VBPI-NF as a first step to empower phylogenetic posterior estimation with deep learning techniques. We first review the PSP parameterization in VBPI that provides the base branch length distribution for our approach. We then develop permutation equivariant normalizing flows for more expressive approximating distributions over the non-Euclidean branch length space across tree topologies, and describe how to incorporate these flows into the VBPI framework for efficient training of the model parameters.

### 3.1 PSP parameterization and the base distribution

Let $\mathbb{S}_r$ denote the set of splits and $\mathbb{S}_{psp}$ denote the set of PSPs. The PSP parameterization assigns parameters $\boldsymbol{\psi}^\mu, \boldsymbol{\psi}^\sigma$ for each element in $\mathbb{S}_r \cup \mathbb{S}_{psp}$. For each edge $e$ on $\tau$, the associated parameters of

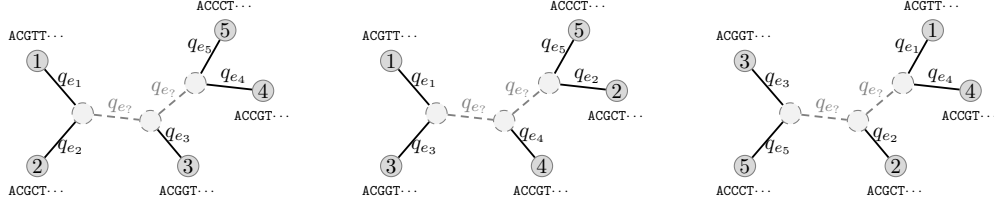

Figure 2: Alignments of the branch length vectors $\boldsymbol{q} = \{q_e\}_{e \in \tau}$ for different tree topologies. The numbers of the tip nodes correspond to the observed species. Other than the branch lengths of the pendant edges that can be aligned according to the associated species of the tip nodes, there is no trivial way to align internal branch lengths for different topologies consistently. However, the pendant (*solid dark*) and internal (*dashed gray*) bipartition of the edges is indeed consistent.

these local structures are then aggregated to form the diagonal Lognormal approximating distribution

$$Q_{\boldsymbol{\psi}}(\boldsymbol{q}|\tau) = \prod_{e \in E(\tau)} p^{\text{Lognormal}}\left(q_e \mid \mu(e, \tau), \sigma(e, \tau)\right)$$

where

$$\mu(e, \tau) = \psi^{\mu}_{e/\tau} + \sum_{s \in e /\!/ \tau} \psi^{\mu}_s, \quad \sigma(e, \tau) = \psi^{\sigma}_{e/\tau} + \sum_{s \in e /\!/ \tau} \psi^{\sigma}_s. \tag{3}$$

In practice, it is usually more convenient to use the log scale branch lengths $\tilde{\boldsymbol{q}} = \log \boldsymbol{q}$ that follow the diagonal Gaussian distribution. We use $\tilde{\boldsymbol{q}}$ in the sequel and refer to them as branch lengths for short.

## 3.2 Permutation equivariant normalizing flows

Although NFs can be powerful when constructing approximating distributions on standard Euclidean space, they are not readily applicable for the non-Euclidean branch length space of phylogenetic models. This is largely due to the lack of a consistent alignment of the branch length vectors (except for the branch lengths of the pendant edges) for different tree topologies (see Figure 2). As a result, it is desirable to have normalizing flows whose transformations of the input variable are, to some extent, permutation equivariant.

**Definition 1** *We say a function $f : \mathbb{R}^d \mapsto \mathbb{R}$ is permutation invariant iff for any permutation $\pi$, $f(x_{\pi(1)}, \cdots, x_{\pi(d)}) = f(x_1, \ldots, x_d)$. We say a transformation $\boldsymbol{f} : \mathbb{R}^d \mapsto \mathbb{R}^d$ is permutation equivariant iff for any permutation $\pi$*

$$\boldsymbol{f}([x_{\pi(1)}, \cdots, x_{\pi(d)}]) = [f_{\pi(1)}(\boldsymbol{x}), \cdots, f_{\pi(d)}(\boldsymbol{x})]. \tag{4}$$

*We say $\boldsymbol{f}$ is permutation equivariant on a subset $S \subset \{1, \ldots, d\}$ iff (4) holds for any permutation $\pi$ of $S$.*

In what follows, we present two types of permutation equivariant normalizing flows for cross topology branch length distributions within the framework of VBPI.

**Permutation equivariant planar flows**   The original planar flows (Rezende and Mohamed, 2015) use planar transformations that are sensitive to the order of the components of the input variable. One can circumvent this problem by assigning the parameters to the components of the input variable as follows. Let $\boldsymbol{x}$ be the input variable, we consider a family of transformations of the following form

$$z_i = x_i + \gamma_{x_i} a\left(\sum_i w_{x_i} x_i + b\right), \quad i = 1, \ldots, d \tag{5}$$

where $\boldsymbol{\gamma} = \{\gamma_{x_i}\}_{i=1}^d$, $\boldsymbol{w} = \{w_{x_i}\}_{i=1}^d$ are the sets of *component-wise* parameters [1] and $a$ is a smooth element-wise nonlinearity. As the parameters permute together with the corresponding components of the input variable, the resulting transformation naturally remains permutation equivariant, and the determinant of its Jacobian can be computed in the same way as standard planar transformations (See section C.1 in the supplement for a proof).

**Proposition 1** *The transformation defined in (5) is permutation equivariant and the determinant of its Jacobian is*

$$\left| \det \frac{\partial \boldsymbol{z}}{\partial \boldsymbol{x}} \right| = \left| 1 + a'(\eta) \sum_i \gamma_{x_i} w_{x_i} \right|, \quad \eta = \sum_i w_{x_i} x_i + b$$

*When $a = \tanh$, the transformation is invertible as long as $\sum_i \gamma_{x_i} w_{x_i} \geq -1$.*

In the context of VBPI, we can assign additional parameters $\boldsymbol{\psi}^\gamma, \boldsymbol{\psi}^w$ for each element in $\mathbb{S}_r \cup \mathbb{S}_{\mathrm{psp}}$ and use the same aggregation for the corresponding parameters of the edges on tree topologies as before. This leads to a permutation equivariant planar transformation of the branch lengths

$$z_e = \tilde{q}_e + \gamma_e a \left( \sum_{e' \in E(\tau)} w_{e'} \tilde{q}_{e'} + b \right), \quad \forall \, e \in E(\tau) \tag{6}$$

where

$$\gamma_e = \psi_{e/\tau}^\gamma + \sum_{s \in e /\!/ \tau} \psi_s^\gamma, \quad w_e = \psi_{e/\tau}^w + \sum_{s \in e /\!/ \tau} \psi_s^w. \tag{7}$$

**Permutation equivariant RealNVP**   RealNVP (Dinh et al., 2017a) provides a family of more expressive normalizing flows that also allow efficiently computable determinants. In RealNVP, the affine coupling transformation scales and shifts one subset $S \subset \{1, \ldots, d\}$ of the $d$ components of the input variable $\boldsymbol{x}$, given the rest $S^c$, as follows

$$
\begin{aligned}
z_i &= x_i, \quad i \in S^c \\
z_i &= x_i \exp\left(\alpha_i(\boldsymbol{x}_{S^c})\right) + \beta_i\left(\boldsymbol{x}_{S^c}\right), \quad i \in S
\end{aligned}
\tag{8}
$$

where $\alpha_i, \beta_i$ are differentiable scalar functions parameterized by neural networks. As the Jacobian of this transformation is lower triangular, its determinant therefore can be easily evaluated.

Now we propose a modification of (8) for cross topology branch length distributions in VBPI-NF. Note that we can naturally partition the edges into the pendant edges and the internal edges, and this bipartition is consistent across tree topologies (see Figure 2). Let $S$ and $S^c$ be such a bipartition. Due to the loss of consistent indices of the branch lengths across tree topologies, we need a transformation that is permutation equivariant on $S$ and $S^c$. Since the identity mapping in (8) is already permutation equivariant, it suffices to require $\alpha_i, \beta_i$ to be permutation invariant of $\boldsymbol{x}_{S^c}$ and permute together with $x_i$. To do this, we assign additional parameters $\boldsymbol{\psi}^w, \boldsymbol{\psi}^{w_\alpha}, \boldsymbol{\psi}^{w_\beta}, \boldsymbol{\psi}^{b_\alpha}, \boldsymbol{\psi}^{b_\beta}$ for each element in $\mathbb{S}_r \cup \mathbb{S}_{\mathrm{psp}}$ as before, and define the following affine coupling transformation for $\tilde{\boldsymbol{q}}$

$$
\begin{aligned}
z_e &= \tilde{q}_e, \quad e \in S^c \\
z_e &= \tilde{q}_e \exp\left(\alpha_e(\tilde{\boldsymbol{q}}_{S^c})\right) + \beta_e(\tilde{\boldsymbol{q}}_{S^c}), \quad e \in S
\end{aligned}
\tag{9}
$$

with

$$
\begin{bmatrix} \alpha_e(\tilde{\boldsymbol{q}}_{S^c}) \\ \beta_e(\tilde{\boldsymbol{q}}_{S^c}) \end{bmatrix} = \begin{bmatrix} (\boldsymbol{w}_e^\alpha)^T \\ (\boldsymbol{w}_e^\beta)^T \end{bmatrix} \rho \left( \sum_{e' \in S^c} \tilde{q}_{e'} \boldsymbol{w}_{e'} + \boldsymbol{b} \right) + \begin{bmatrix} b_e^\alpha \\ b_e^\beta \end{bmatrix} \tag{10}
$$

where the parameters for each edge $e \in E(\tau)$ are obtained via the same aggregation as before

$$
\begin{aligned}
\boldsymbol{w}_e &= \boldsymbol{\psi}_{e/\tau}^{\boldsymbol{w}} + \sum_{s \in e /\!/ \tau} \boldsymbol{\psi}_s^{\boldsymbol{w}}, \quad \boldsymbol{w}_e^\alpha = \boldsymbol{\psi}_{e/\tau}^{\boldsymbol{w}_\alpha} + \sum_{s \in e /\!/ \tau} \boldsymbol{\psi}_s^{\boldsymbol{w}_\alpha}, \quad \boldsymbol{w}_e^\beta = \boldsymbol{\psi}_{e/\tau}^{\boldsymbol{w}_\beta} + \sum_{s \in e /\!/ \tau} \boldsymbol{\psi}_s^{\boldsymbol{w}_\beta} \\
b_e^\alpha &= \psi_{e/\tau}^{b_\alpha} + \sum_{s \in e /\!/ \tau} \psi_s^{b_\alpha}, \quad b_e^\beta = \psi_{e/\tau}^{b_\beta} + \sum_{s \in e /\!/ \tau} \psi_s^{b_\beta}
\end{aligned}
\tag{11}
$$

and $\rho$ is a standard neural network or a simple smooth element-wise nonlinear function. The neural nets proposed in (10) can be viewed as a variant of standard neural nets where weights of the input and output layers permute together with the input variable and hence remain permutation equivariant.

**Proposition 2** *The transformation defined in (9) is permutation equivariant on $S$ and $S^c$.*

A proof of Proposition 2 is provided in section C.2 in the supplement. We can alternate $S$ and $S^c$ when stacking these coupling layers as suggested in Dinh et al. (2017a).

### 3.3 Parameter training in VBPI-NF

Having presented the permutation equivariant normalizing flows for more expressive branch length distributions across tree topologies, we now illustrate how to integrate them into the VBPI framework. Let $\boldsymbol{f}_1, \ldots, \boldsymbol{f}_L$ be the sequence of permutation equivariant transformations in a normalizing flow. Let $Q_{\boldsymbol{\psi}}(\tilde{\boldsymbol{q}}^{(0)}|\tau)$ denote the density function of the base distribution of the branch lengths on a tree topology $\tau$, which is a diagonal Gaussian distribution with PSP parameterization as in section 3.1. The density function of the transformed approximation is given by (2), which is then used for the lower bound computation. Following Zhang and Matsen IV (2019), we use an annealed multi-sample lower bound for VBPI-NF that takes the form (see section D in the supplement for more details)

$$\tilde{L}_{\lambda_n}^K(\boldsymbol{\phi}, \boldsymbol{\psi}, \boldsymbol{\psi}^{\mathrm{NF}}) = \mathbb{E}_{Q_{\boldsymbol{\phi},\boldsymbol{\psi}}\left(\tau^{1:K},(\tilde{\boldsymbol{q}}^{(0)})^{1:K}\right)} \log\left(\frac{1}{K}\sum_{i=1}^{K}\frac{\left[p\left(Y|\tau^i,\left(\tilde{\boldsymbol{q}}^{(L+1)}\right)^i\right)\right]^{\lambda_n} p\left(\tau^i,\left(\tilde{\boldsymbol{q}}^{(L+1)}\right)^i\right)}{Q_{\boldsymbol{\phi}}(\tau^i)Q_{\boldsymbol{\psi}}\left(\left(\tilde{\boldsymbol{q}}^{(0)}\right)^i|\tau^i\right)\prod_{\ell=0}^{L}\left|\det\frac{\partial\left(\tilde{\boldsymbol{q}}^{(\ell+1)}\right)^i}{\partial\left(\tilde{\boldsymbol{q}}^{(\ell)}\right)^i}\right|^{-1}}\right)$$

where $\tilde{\boldsymbol{q}}^{(\ell+1)} = \boldsymbol{f}_{\ell+1}(\tilde{\boldsymbol{q}}^{(\ell)})$ for $\ell = 0, \ldots, L-1$, and $\tilde{\boldsymbol{q}}^{(L+1)} = \exp\left(\tilde{\boldsymbol{q}}^{(L)}\right)$;[2] $\left(\tilde{\boldsymbol{q}}^{(\ell)}\right)^i$ means the $i$-th sample of $\tilde{\boldsymbol{q}}^{(\ell)}$, $i = 1, \ldots, K$, $\ell = 0, \ldots, L+1$; $\boldsymbol{\psi}^{\mathrm{NF}}$ denotes the flow parameters; $\lambda_n$ is the inverse temperature at the $n$-th iteration that follows an annealing schedule (see section 5 for an example). With efficient Monte Carlo gradient estimates, the above lower bound can be maximized the same way as in Zhang and Matsen IV (2019). See algorithm 1 in the supplement for more details.

## 4 Related work

Zaheer et al. (2017) proposed a permutation invariant architecture over sets, called DeepSets, where permutation invariance is achieved by performing a permutation invariant *pooling* operation (e.g., a sum) after mapping each element in a set into a learned feature representation via a shared feed-forward neural network. Thanks to the compact structural representation (e.g., splits and PSPs) of the edges of phylogenetic tree topologies, our permutation invariant/equivariant architectures for the branch lengths (6, 10) allow each element to have its own set of parameters, and hence could provide more flexible approximations. Liu et al. (2019) and Bender et al. (2020) extended normalizing flows to unordered data (e.g., graphs, point clouds, etc). However, their flow-based models require splitting the feature vector of each data point and thus are only suitable for modeling distributions over sets of vectors. In contrast, our permutation equivariant normalizing flows are for the branch lengths of phylogenetic trees, which are sets of scalars. Köhler et al. (2020) and Rezende et al. (2019) described methods for building equivariant transformations in the context of continuous-time flows.

## 5 Experiments

In this section, we compare VBPI-NF to VBPI on two benchmark tasks for Bayesian phylogenetic inference: posterior approximation and marginal likelihood estimation. Following Zhang and Matsen IV (2019), we use the simplest SBN for the tree topology variational distribution, and estimate the CPT supports from ultrafast maximum likelihood phylogenetic bootstrap trees using UFBoot (Minh et al., 2013). We use VBPI with PSP branch length parameterization as our baseline and refer to it as PSP in the sequel. The code is available at `https://github.com/zcrabbit/vbpi-nf`.

### 5.1 Datasets and experimental setup

We performed experiments on 8 real datasets that are commonly used to benchmark Bayesian phylogenetic inference methods (Hedges et al., 1990; Garey et al., 1996; Yang and Yoder, 2003; Henk et al., 2003; Lakner et al., 2008; Zhang and Blackwell, 2001; Yoder and Yang, 2004; Rossman et al., 2001; Höhna and Drummond, 2012; Larget, 2013; Whidden and Matsen IV, 2015). These datasets, which we will call DS1-8, consist of sequences from 27 to 64 eukaryote species with 378 to 2520 site observations (see Table 1 and Lakner et al. (2008)). As in Zhang and Matsen IV

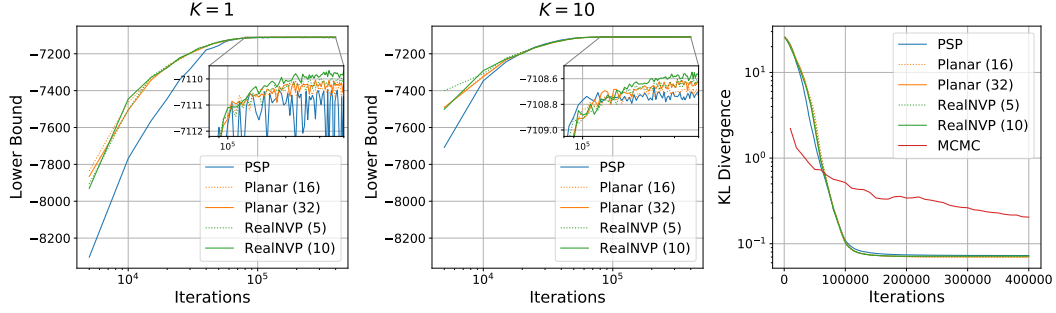

Figure 3: Performance on DS1. **Left & Middle:** Lower bounds ($K = 1, 10$). **Right:** KL divergence to the ground truth posterior over tree topologies. The number in brackets specifies the number of layers used in different flow models. MCMC results are averaged over 10 independent runs.

Table 1: Lower bound (LB) and marginal likelihood (ML) estimates of different methods across 8 benchmark datasets for Bayesian phylogenetic inference. The marginal likelihood estimates of all variational methods are obtained via importance sampling using 1000 samples, and the results (in units of nats) are averaged over 100 independent runs with standard deviation in brackets. Results for stepping-stone (SS) are from Zhang and Matsen IV (2019) (using 10 independent MrBayes (Ronquist et al., 2012) runs, each with 4 chains for 10,000,000 iterations and sampled every 100 iterations).

|  | DATA SET | DS1 | DS2 | DS3 | DS4 | DS5 | DS6 | DS7 | DS8 |
|---|---|---|---|---|---|---|---|---|---|
|  | # TAXA | 27 | 29 | 36 | 41 | 50 | 50 | 59 | 64 |
|  | # SITES | 1949 | 2520 | 1812 | 1137 | 378 | 1133 | 1824 | 1008 |
| LB (K=1) | PSP | -7111.23(1.04) | -26369.63(0.69) | -33736.60(0.33) | -13332.37(0.54) | -8218.35(0.20) | -6729.27(0.50) | -37335.15(0.11) | -8655.48(0.38) |
|  | PLANAR(16) | -7110.33(0.16) | -26368.80(0.27) | -33736.14(0.14) | -13331.92(0.11) | -8217.98(0.13) | -6728.89(0.18) | -37334.78(0.11) | -8655.15(0.17) |
|  | PLANAR(32) | -7110.22(0.17) | -26368.69(0.23) | -33736.02(0.21) | -13331.73(0.12) | -8217.90(0.14) | -6728.68(0.19) | -37334.60(0.12) | -8654.97(0.16) |
|  | REALNVP(5) | -7110.12(0.13) | -26368.75(0.24) | -33735.86(0.10) | -13331.71(0.11) | -8217.80(0.14) | -6728.54(0.15) | -37334.44(0.11) | -8654.62(0.13) |
|  | REALNVP(10) | **-7109.80(0.11)** | **-26368.59(0.23)** | **-33735.81(0.12)** | **-13331.39(0.08)** | **-8217.56(0.12)** | **-6728.04(0.14)** | **-37333.94(0.09)** | **-8654.02(0.12)** |
| LB (K=10) | PSP | -7108.73(0.02) | -26367.88(0.02) | -33735.29(0.02) | -13330.34(0.03) | -8215.57(0.04) | -6725.48(0.04) | -37332.69(0.03) | -8651.88(0.04) |
|  | PLANAR(16) | -7108.70(0.02) | -26367.80(0.01) | -33735.21(0.01) | -13330.28(0.02) | -8215.44(0.04) | -6725.42(0.04) | -37332.50(0.03) | -8651.80(0.04) |
|  | PLANAR(32) | -7108.64(0.02) | -26367.77(0.01) | -33735.17(0.01) | -13330.22(0.02) | -8215.37(0.03) | -6725.32(0.04) | -37332.43(0.03) | -8651.72(0.04) |
|  | REALNVP(5) | -7108.63(0.02) | -26367.77(0.01) | -33735.18(0.01) | -13330.22(0.02) | -8215.36(0.03) | -6725.33(0.04) | -37332.42(0.03) | -8651.62(0.04) |
|  | REALNVP(10) | **-7108.58(0.02)** | **-26367.75(0.01)** | **-33735.16(0.01)** | **-13330.16(0.02)** | **-8215.29(0.03)** | **-6725.18(0.04)** | **-37332.30(0.02)** | **-8651.41(0.03)** |
| ML | PSP | -7108.39(0.18) | -26367.71(0.08) | -33735.09(0.10) | -13329.93(0.21) | -8214.44(0.48) | -6724.13(0.48) | -37331.92(0.32) | -8650.12(0.58) |
|  | PLANAR(16) | -7108.39(0.15) | -26367.70(0.07) | -33735.09(0.07) | -13329.93(0.17) | -8214.49(0.42) | -6724.25(0.45) | -37331.91(0.26) | -8650.42(0.52) |
|  | PLANAR(32) | -7108.40(0.14) | -26367.70(0.06) | **-33735.09(0.05)** | -13329.93(0.16) | -8214.50(0.38) | -6724.19(0.44) | -37331.93(0.23) | -8650.40(0.50) |
|  | REALNVP(5) | -7108.40(0.14) | **-26367.71(0.04)** | -33735.09(0.06) | -13329.92(0.16) | -8214.50(0.38) | -6724.28(0.39) | **-37331.92(0.22)** | -8650.46(0.44) |
|  | REALNVP(10) | **-7108.39(0.11)** | **-26367.71(0.04)** | **-33735.09(0.05)** | **-13329.92(0.13)** | -8214.51(0.36) | **-6724.25(0.37)** | -37331.90(0.22) | **-8650.42(0.41)** |
|  | SS | -7108.42(0.18) | -26367.57(0.48) | -33735.44(0.50) | -13330.06(0.54) | **-8214.51(0.28)** | -6724.07(0.86) | -37332.76(2.42) | -8649.88(1.75) |

(2019), we concentrate on the most challenging part of the phylogenetic model: joint learning of the tree topologies and the branch lengths, and assume the other components are given as follows: a uniform prior on the tree topology, an i.i.d. exponential prior ($\mathrm{Exp}(10)$) for the branch lengths and the simple Jukes and Cantor (1969) substitution model that often produces difficult tree posteriors. We gather the support of CPTs from 10 replicates of 10000 ultrafast maximum likelihood bootstrap trees (Minh et al., 2013). We set $K = 10$ for the multi-sample lower bound, with a schedule $\lambda_n = \min(1, 0.001 + n/100000)$, going from 0.001 to 1 after 100000 iterations. We evaluate the performance of our permutation equivariant normalizing flows with varying numbers of layers. All models were implemented in Pytorch (Paszke et al., 2019) with the Adam optimizer (Kingma and Ba, 2015), where the Monte Carlo gradient estimates for the tree topology parameters and branch length parameters were obtained via VIMCO (Mnih and Rezende, 2016) and the reparameterization trick (Kingma and Welling, 2014) respectively as in Zhang and Matsen IV (2019). We designed our experiments with the goals of (i) verifying the improvement of VBPI-NF over the baseline method and (ii) investigating how different components of normalizing flows are combined for the overall improvement of variational approximations. Results were collected after 400,000 parameter updates.

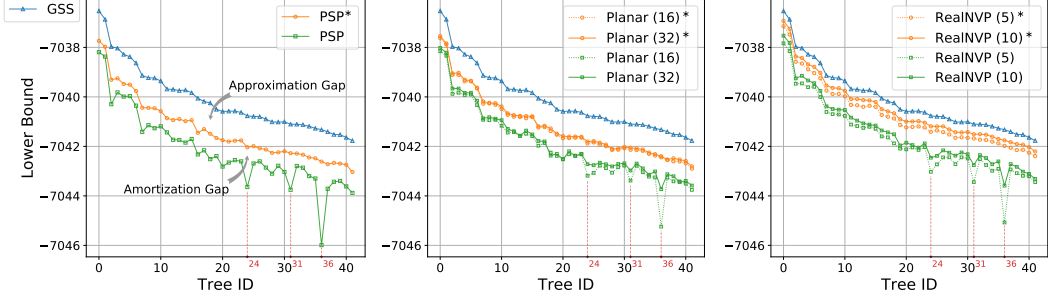

Figure 4: Performance on trees in the 95% credible set of DS1. The star $^*$ indicates the best lower bounds for the corresponding family of approximating distributions on each tree topology. **Left:** Lower bounds and inference gaps for PSP. **Middle:** Lower bounds for Planar flows. **Right:** Lower bounds for RealNVP. The number in brackets specifies the number of layers used in different flow models. All lower bounds were computed by averaging over 10000 Monte Carlo samples.

## 5.2    Results

**Lower bound and marginal likelihood estimation**    Table 1 shows the estimates of the lower bounds ($K = 1, 10$) and the marginal likelihood from different variational approaches on the 8 benchmark datasets. Results for the stepping-stone (SS) method (Xie et al., 2011) (one of the state-of-the-art sampling based methods for marginal likelihood estimation) from Zhang and Matsen IV (2019) are also reported. For each data set $\boldsymbol{Y}$, the marginal likelihood estimates provided by different methods are for the same marginal likelihood $\log p(\boldsymbol{Y})$, and better approximation leads to lower variance. The numbers of layers in the flow models are in parentheses. We see that VBPI-NF using permutation equivariant normalizing flows consistently outperform the PSP baseline, with more expressive flows achieving tighter lower bounds. Moreover, when used as importance distributions for marginal likelihood estimation via importance sampling, variational approaches compare favorably to SS with much fewer samples. Thanks to the flexible normalizing flows, VBPI-NF provides more steady estimates (much less variance) than PSP, and therefore would be more reliable for downstream tasks such as Bayesian model selection. Figure 3 shows the lower bounds ($K = 1, 10$) and KL divergence to the ground truth posterior over tree topologies [3] as a function of the number of parameter updates on DS1. Although flows tend to converge slower than standard approaches in VI, we see that by the time PSP converges, the proposed flow-based methods achieve comparable (if not better) lower bounds and quickly surpass PSP as the number of iterations increases. When compared to MCMC in terms of the KL divergence to the ground truth tree topology posterior using similar settings as in Zhang and Matsen IV (2019), flow-based variational methods maintain the speed advantage of PSP and arrive at good approximations with much less computation.

**Lower bounds and inference gaps on tree topologies**    To better understand the effect of normalizing flows for the overall approximation improvement, we further investigate the approximation performance of different methods on individual trees in the 95% credible set of DS1. For a tree topology $\tau$, we define the lower bound $L(Q_{\boldsymbol{\psi}}|\tau)$ of an approximating distribution $Q_{\boldsymbol{\psi}}(\boldsymbol{q}|\tau)$, and the maximum lower bound $L(Q^*|\tau)$ that can be achieved by distributions from the corresponding approximating family $\mathcal{Q}$ as follows

$$L(Q_{\boldsymbol{\psi}}|\tau) = \mathbb{E}_{Q_{\boldsymbol{\psi}}(\boldsymbol{q}|\tau)} \log \left( \frac{p(\boldsymbol{Y}|\tau, \boldsymbol{q})p(\boldsymbol{q})}{Q_{\boldsymbol{\psi}}(\boldsymbol{q}|\tau)} \right) \leq \log p(\boldsymbol{Y}|\tau), \quad L(Q^*|\tau) = \max_{Q_{\boldsymbol{\psi}} \in \mathcal{Q}} L(Q_{\boldsymbol{\psi}}|\tau).$$

We then follow Cremer et al. (2018) to break the inference gap of the approximating distribution on $\tau$, which is the difference between the marginal log-likelihood $\log p(\boldsymbol{Y}|\tau)$ and the lower bound $L(Q_{\boldsymbol{\psi}}|\tau)$, into two components, i.e. the approximation gap $\log p(\boldsymbol{Y}|\tau) - L(Q^*|\tau)$ and the amortization gap $L(Q^*|\tau) - L(Q_{\boldsymbol{\psi}}|\tau)$. Figure 4 shows the lower bounds given by different approximating distributions obtained from PSP and VBPI-NF, together with the corresponding maximum lower bounds $L(Q^*|\tau)$,

Table 2: Inference gaps on trees in the 95% credible set of DS1. The ALL columns refer to the average gaps over all trees in the credible set.

| GAP | PSP | | PLANAR (16) | | PLANAR (32) | | REALNVP (5) | | REALNVP (10) | |
|---|---|---|---|---|---|---|---|---|---|---|
| | TREE 36 | ALL | TREE 36 | ALL | TREE 36 | ALL | TREE 36 | ALL | TREE 36 | ALL |
| APPROXIMATION | 1.29 | 1.21 | 1.12 | 1.08 | 1.07 | 1.02 | 0.65 | 0.62 | **0.43** | **0.40** |
| AMORTIZATION | 3.37 | 0.84 | 2.80 | 0.82 | **1.33** | **0.72** | 3.10 | 0.98 | 1.83 | 0.93 |
| INFERENCE | 4.66 | 2.05 | 3.92 | 1.90 | 2.40 | 1.74 | 3.75 | 1.60 | **2.26** | **1.33** |

on each tree topology $\tau$. The ground truth marginal log-likelihoods $\log p(\boldsymbol{Y}|\tau)$ for each tree topology $\tau$ are also reported, which were estimated using the state-of-the-art generalized stepping-stone (GSS) algorithm (Fan et al., 2011). The left plot in Figure 4 shows the results for PSP. We see there is a large approximation gap, indicating that distributions that completely ignore the correlation among the parameters such as the diagonal Lognormal distribution used by PSP may be too restrictive to fit the true branch length posteriors well. Moreover, the simple amortization in PSP was not able to generalize well in the tree topology space, as evidenced by the significant performance drop on certain tree topologies (e.g., tree 24, 31, 36). The middle and right plots present the results using the permutation equivariant normalizing flows in VBPI-NF. As expected, we see the approximation gaps for normalizing flows (especially RealNVP) are considerably smaller than those for PSP, showing the effectiveness of flow-based distributions for phylogenetic posterior approximations. With carefully designed permutation equivariant architectures, the extra flexibility of normalizing flows managed to generalize in the tree topology space, resulting in an overall lower bound improvement over PSP across different tree topologies. More interestingly, we see that using more layers in normalizing flows not only helps to reduce the approximation gaps, but also helps to reduce the amortization gaps, especially on those challenging tree topologies for PSP. This indicates the parameters used for improving the expressiveness of the approximation may also play a role in reducing the amortization error (as observed in Cremer et al. (2018)), which is partially due to the structured parameterization (7, 11) in our permutation equivariant normalizing flows that incorporates local topological information into each layer. More detailed results of the approximation gaps, amortization gaps and inference gaps are summarized in Table 2 which validate our observations. Note that as the expressiveness of the approximating distributions increases, the amortization gap could become more significant compared to the approximation gap. Designing more flexible amortization architectures over phylogenetic tree topologies, therefore, would be an interesting subject that we leave to future work.

## 6 Conclusion

We introduced VBPI-NF, a new type of variational Bayesian phylogenetic inference method that leverages flexible normalizing flows for more expressive branch length approximations. By handling the non-Euclidean branch length space of phylogenetic models with carefully designed permutation equivariant transformations, normalizing flows in VBPI-NF provide a rich family of flexible branch length distributions that generalize across different tree topologies. In experiments, we demonstrated that VBPI-NF consistently outperforms the baseline approach on a benchmark of real data Bayesian phylogenetic inference tasks. A number of extensions and potential improvements are possible, such as incorporating more powerful normalizing flows, designing more flexible amortization architectures over tree topologies and extending VBPI-NF to infer rooted, time-measured phylogenetic trees.

## Broader Impact

Bayesian phylogenetic inference has been applied to a wide range of applications, including genomic epidemiology, conservation genetics, comparative immunology, vaccine design and many more. Our research could be used as a faster alternative to the current MCMC based Bayesian phylogenetic inference methods used in these applications, to deliver reasonably accurate posterior estimates in a more timely manner. Moreover, our use of normalizing flows for more expressive branch length approximations demonstrates the power of deep Bayesian learning for models with complex, highly

structured, non-Euclidean parameter space and is likely to drive the development of new deep learning methods for phylogenetic models and other models with similar structures.

## Acknowledgments and Disclosure of Funding

This work was supported by the Key Laboratory of Mathematics and Its Applications (LMAM) and the Key Laboratory of Mathematical Economics and Quantitative Finance (LMEQF) of Peking University. The author is grateful for the computational resources provided by the High-performance Computing Platform of Peking University. The author appreciates the anonymous NeurIPS reviewers for their constructive feedback. The author would also like to thank the Matsen group at Fred Hutch, where he started research in the field of Bayesian phylogenetic inference.

## Footnotes

[1]The component-wise parameters $\gamma_{x_i}, w_{x_i}$ are associated with (hence permute together with) the input variable $x_i$. They do not depend on the value of $x_i$.

[2]The last exponential transformation is to map the branch lengths back to the non-negative domain. When $L = 0$, this lower bound reduces to the annealed version of the VBPI lower bound (1).

[3]As in Zhang and Matsen IV (2019), the ground truth was obtained from an extremely long MCMC run of 10 billion iterations (sampled each 1000 iterations with first 25% discarded as burn-in) using MrBayes.

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
