[Supplementary Material]

# Supplement for Improved Variational Bayesian Phylogenetic Inference with Normalizing Flows

## A   Subsplit Bayesian networks

Figure 1: A simple subsplit Bayesian network for a leaf set that contains 4 species A, B, C and D. **Left**: Examples of rooted phylogenetic trees. **Middle**: The corresponding SBN assignments. For ease of illustration, subsplit $(W, Z)$ is represented as $\frac{W}{Z}$ in the graph. The *dashed gray subgraphs* represent fake splitting processes where splits are deterministically assigned, and are used purely to complement the networks such that the overall network has a fixed structure. **Right**: The SBN for these examples. This figure is adapted from Zhang and Matsen IV (2019).

Subsplit Bayesian networks (SBNs) introduced by Zhang and Matsen IV (2018) provide a family of flexible distributions on tree topologies. A subsplit Bayesian network $B_{\mathcal{X}}$ on a leaf set $\mathcal{X}$ of size $N$ is a Bayesian network where the nodes take on subsplit or singleton clade values that represent the local topological structures of trees (Figure 1). To encode a rooted tree topology to an SBN representation, one can follow the splitting process (see the *solid dark subgraphs* in Figure 1, middle) of the tree and assign the subsplits to the corresponding nodes along the way, resulting in a unique subsplit decomposition of the tree topology. Given the subsplit decomposition of a rooted tree $\tau = \{s_1, s_2, \dots\}$, where $s_1$ is the root subsplit, the SBN-induced tree probability of $\tau$ is

$$p_{\text{sbn}}(T = \tau) = p(S_1 = s_1) \prod_{i>1} p(S_i = s_i | S_{\pi_i} = s_{\pi_i})$$

where $S_i$ denote the subsplit- or singleton-clade-valued random variables at node $i$ and $\pi_i$ is the index set of the parents of $S_i$. As Bayesian networks, SBN-induced distributions are all naturally normalized. We can also adjust the structures of SBNs for a wide range of expressive distributions, as long as they remain valid directed acyclic graphs (DAGs). Although in practice, we find the simplest SBN (the one with a full and complete binary tree structure as shown in Figure 1) is good enough.

The SBN framework also generalizes to unrooted trees, which are the most common type of phylogenetic trees. By viewing unrooted trees as rooted trees with unobserved roots and marginalizing out the unobserved root node, we have the SBN probability estimates for unrooted trees

$$p_{\text{sbn}}(T^{\text{u}} = \tau) = \sum_{s_1 \sim \tau} p(S_1 = s_1) \prod_{i>1} p(S_i = s_i | S_{\pi_i} = s_{\pi_i})$$

where $\sim$ means all root subsplits that are compatible with $\tau$ (i.e., root subsplits of the edges of $\tau$).

## B   More details on variational Bayesian phylogenetic inference

The family of approximating distributions used in variational Bayesian phylogenetic inference (VBPI) is formed as $Q_{\phi,\psi} = Q_{\phi}(\tau) \cdot Q_{\psi}(q|\tau)$, which is the product of an SBN-based distribution $Q_{\phi}(\tau)$

over the tree topologies and a diagonal Lognormal distribution $Q_{\psi}(\boldsymbol{q}|\tau)$ over the branch lengths. The best approximation is obtained by maximizing the multi-sample lower bound

$$\boldsymbol{\phi}^*, \boldsymbol{\psi}^* = \underset{\boldsymbol{\phi},\boldsymbol{\psi}}{\arg\min}\, \mathbb{E}_{Q_{\boldsymbol{\phi},\boldsymbol{\psi}}(\tau^{1:K},\boldsymbol{q}^{1:K})} \log\left(\frac{1}{K}\sum_{i=1}^{K}\frac{p(\boldsymbol{Y}|\tau^i,\boldsymbol{q}^i)p(\tau^i,\boldsymbol{q}^i)}{Q_{\boldsymbol{\phi}}(\tau^i)Q_{\boldsymbol{\psi}}(\boldsymbol{q}^i|\tau^i)}\right)$$

where $Q_{\boldsymbol{\phi},\boldsymbol{\psi}}(\tau^{1:K},\boldsymbol{q}^{1:K}) = \prod_{i=1}^{K} Q_{\boldsymbol{\phi}}(\tau^i)Q_{\boldsymbol{\psi}}(\boldsymbol{q}^i|\tau^i)$. To parameterize SBNs in VBPI, we need a sufficiently large subsplit *support* of CPTs (i.e., where the associate conditional probabilities are allowed to take nonzero values) that covers favorable parent child subsplit pairs from trees with high posterior probabilities. In practice, a simple bootstrap-based approach has been found effective for providing such a support (Zhang and Matsen IV, 2019). Let $\mathbb{S}_r$ denote the set of root subsplits (e.g., the splits) in the support and $\mathbb{S}_{\mathrm{ch|pa}}$ denote the set of parent-child subsplit pairs in the support. The CPTs can be defined via the softmax function as follows

$$p(S_1 = s_1) = \frac{\exp(\phi_{s_1})}{\sum_{s_r \in \mathbb{S}_r} \exp(\phi_{s_r})}, \quad p(S_i = s|S_{\pi_i} = t) = \frac{\exp(\phi_{s|t})}{\sum_{s \in \mathbb{S}_{\cdot|t}} \exp(\phi_{s|t})}$$

We can evaluate the SBN probabilities of tree topologies efficiently through a two pass algorithm (Zhang and Matsen IV, 2018). Sampling from SBNs is also straightforward via ancestral sampling.

As the naive brute-force parameterization for the branch length distributions of different tree topologies requires a large number of parameters when the high-probability domain of the tree topology posterior are diffuse, Zhang and Matsen IV (2019) amortized the branch length variational distribution over different tree topologies via their shared local structures. For example, one can simply use the splits of the edges on phylogenetic trees, and assign parameters for each split in $\mathbb{S}_r$. A more sophisticated parameterization that uses more tree-dependent information, i.e., primary subsplit pairs (PSPs), has been found to provide better approximations across tree topologies.

## C   Proofs for permutation equivariance

### C.1   Proof of proposition 1

*Proof.* For any permutation $\pi$, we have

$$z_{\pi(i)} = x_{\pi(i)} + \gamma_{x_{\pi(i)}} a\left(\sum_i w_{x_{\pi(i)}} x_{\pi(i)} + b\right) = x_{\pi(i)} + \gamma_{x_{\pi(i)}} a\left(\sum_i w_{x_i} x_i + b\right).$$

Therefore, transformation in (5) is permutation equivariant. Let $\eta = \sum_i w_{x_i} x_i + b$,

$$\frac{\partial \boldsymbol{z}}{\partial \boldsymbol{x}} = \boldsymbol{I} + a'(\eta)\boldsymbol{\gamma_x}\boldsymbol{w}^T \Rightarrow \left|\det \frac{\partial \boldsymbol{z}}{\partial \boldsymbol{x}}\right| = \left|\det(\boldsymbol{I} + a'(\eta)\boldsymbol{\gamma_x}\boldsymbol{w}^T)\right| = \left|1 + a'(\eta)\sum_i \gamma_{x_i} w_{x_i}\right|$$

When $a = \tanh$, $0 < a'(\eta) < 1$. Therefore, the transformation is invertible if $\sum_i \gamma_{x_i} w_{x_i} \geq -1$. To satisfy this condition, we can use the same numerically stable parameterization as in Rezende and Mohamed (2015). Note that the determinant of the Jacobian is permutation invariant.   □

### C.2   Proof of proposition 2

*Proof.* Let $\pi$ be a permutation of $S$ and $S^c$, that is $\pi(S)$ is a rearrangement of $S$ and $\pi(S^c)$ is a rearrangement of $S^c$. Since the affine coupling transformation in (9) keeps $S^c$ untouched, we have

$$z_{\pi(e)} = \tilde{q}_{\pi(e)}, \quad \forall\, e \in S^c$$

and $\forall\, e \in S$,

$$z_{\pi(e)} = \tilde{q}_{\pi(e)} \exp\left(\alpha_{\pi(e)}(\tilde{\boldsymbol{q}}_{\pi(S^c)})\right) + \beta_{\pi(e)}(\tilde{\boldsymbol{q}}_{\pi(S^c)})$$
$$= \tilde{q}_{\pi(e)} \exp\left(\alpha_{\pi(e)}(\tilde{\boldsymbol{q}}_{S^c})\right) + \beta_{\pi(e)}(\tilde{\boldsymbol{q}}_{S^c}).$$

The last equality is due to the permutation invariance of $\alpha_{\pi(e)}$ and $\beta_{\pi(e)}$ on $S^c$, which can be easily verified as follows[1]

$$\alpha_{\pi(e)}(\tilde{\boldsymbol{q}}_{\pi(S^c)}) = (\boldsymbol{w}_{\pi(e)}^{\alpha})^T \rho\left(\sum_{e' \in \pi(S^c)} \tilde{q}_{e'}\boldsymbol{w}_{e'} + \boldsymbol{b}\right) = (\boldsymbol{w}_{\pi(e)}^{\alpha})^T \rho\left(\sum_{e' \in S^c} \tilde{q}_{e'}\boldsymbol{w}_{e'} + \boldsymbol{b}\right) = \alpha_{\pi(e)}(\tilde{\boldsymbol{q}}_{S^c})$$

Therefore, the transformation in (9) is permutation equivariant on $S$ and $S^c$. $\qquad\square$

## D   The lower bound for VBPI-NF

Let $\bar{\psi} = (\psi, \psi^{\text{NF}})$. By the change of variable formula (2), the density of the transformed branch length approximation in VBPI-NF is

$$Q_{\bar{\psi}}(\tilde{\boldsymbol{q}}^{(L+1)}|\tau) = Q_{\psi}(\tilde{\boldsymbol{q}}^{(0)}|\tau) \prod_{\ell=0}^{L} \left| \det \frac{\partial \tilde{\boldsymbol{q}}^{(\ell+1)}}{\partial \tilde{\boldsymbol{q}}^{(\ell)}} \right|^{-1} \tag{D.1}$$

where $Q_{\psi}(\tilde{\boldsymbol{q}}^{(0)}|\tau)$ is the density function of a diagonal Gaussian distribution and the last iterate $\tilde{\boldsymbol{q}}^{(L+1)} = \exp(\tilde{\boldsymbol{q}}^{(L)})$ maps the branch lengths back to the non-negative domain. The approximating distribution in VBPI-NF then takes the following form

$$Q_{\phi,\bar{\psi}}(\boldsymbol{q},\tau) = Q_{\phi}(\tau)Q_{\bar{\psi}}(\boldsymbol{q}|\tau)$$

and we can compute the annealed version of the multi-sample lower bound (Burda et al., 2016; Mnih and Rezende, 2016) as follows

$$\tilde{L}_{\lambda_n}^K(\boldsymbol{\phi}, \boldsymbol{\psi}, \boldsymbol{\psi}^{\text{NF}}) = \mathbb{E}_{Q_{\phi,\bar{\psi}}\left(\tau^{1:K}, (\tilde{\boldsymbol{q}}^{(L+1)})^{1:K}\right)} \log \left( \frac{1}{K} \sum_{i=1}^{K} \frac{\left[ p\left(Y|\tau^i, (\tilde{\boldsymbol{q}}^{(L+1)})^i\right)\right]^{\lambda_n} p\left(\tau^i, (\tilde{\boldsymbol{q}}^{(L+1)})^i\right)}{Q_{\phi}(\tau^i) Q_{\bar{\psi}}\left((\tilde{\boldsymbol{q}}^{(L+1)})^i|\tau^i\right)} \right)$$

$$= \mathbb{E}_{Q_{\phi,\psi}\left(\tau^{1:K}, (\tilde{\boldsymbol{q}}^{(0)})^{1:K}\right)} \log \left( \frac{1}{K} \sum_{i=1}^{K} \frac{\left[ p\left(Y|\tau^i, (\tilde{\boldsymbol{q}}^{(L+1)})^i\right)\right]^{\lambda_n} p\left(\tau^i, (\tilde{\boldsymbol{q}}^{(L+1)})^i\right)}{Q_{\phi}(\tau^i) Q_{\psi}\left((\tilde{\boldsymbol{q}}^{(0)})^i|\tau^i\right) \prod_{\ell=0}^{L} \left| \det \frac{\partial (\tilde{\boldsymbol{q}}^{(\ell+1)})^i}{\partial (\tilde{\boldsymbol{q}}^{(\ell)})^i} \right|^{-1}} \right)$$

The last equation is due to the law of the unconscious statistician (LOTUS). When $L = 0$ (no normalizing flows involved), $\tilde{\boldsymbol{q}}^{(1)} = \exp(\tilde{\boldsymbol{q}}^{(0)})$ follows the diagonal Lognormal distribution. Therefore, the density in (D.1) is just the density function of the diagonal Lognormal distribution and the above annealed multi-sample lower bound for VBPI-NF reduces to the annealed multi-sample lower bound for VBPI (Zhang and Matsen IV, 2019).

## E   The VBPI-NF Alogrithm

---
**Algorithm 1** The VBPI-NF algorithm
---
1: $\boldsymbol{\phi}, \boldsymbol{\psi}, \boldsymbol{\psi}^{\text{NF}} \leftarrow$ Initialize parameters, $n = 1$
2: **while** not converged **do**
3: $\quad \tau^1, \ldots, \tau^K \leftarrow$ Random samples from the current SBN-based tree space approximating distribution $Q_{\phi}(\tau)$ via ancestral sampling
4: $\quad \boldsymbol{\epsilon}^1, \ldots, \boldsymbol{\epsilon}^K \leftarrow$ Random samples from the multivariate standard normal distribution $\mathcal{N}(\boldsymbol{0}, \boldsymbol{I})$
5: $\quad \boldsymbol{g} \leftarrow \nabla_{\boldsymbol{\phi}, \boldsymbol{\psi}, \boldsymbol{\psi}^{\text{NF}}} \tilde{L}_{\lambda_n}^K(\boldsymbol{\phi}, \boldsymbol{\psi}, \boldsymbol{\psi}^{\text{NF}}; \tau^{1:K}, \boldsymbol{\epsilon}^{1:K})$ (Use any suitable Monte Carlo gradient estimate, see Zhang and Matsen IV (2019) for examples)
6: $\quad \boldsymbol{\phi}, \boldsymbol{\psi}, \boldsymbol{\psi}^{\text{NF}} \leftarrow$ Update parameters using gradients $\boldsymbol{g}$ (e.g., SGA)
7: $\quad n \leftarrow n + 1$
8: **end while**
9: **return** $\boldsymbol{\phi}, \boldsymbol{\psi}, \boldsymbol{\psi}^{\text{NF}}$
---

## Footnotes

[1]We show the case of $\alpha_{\pi(e)}$ here, and $\beta_{\pi(e)}$ follows similarly.