[Reviews · NeurIPS 2020]

Review 1

Summary and Contributions: This paper expands the variational family for the branch length distributions (in phylogenetic tree) using normalizing flow (NF) for tree approximation. Their main contribution over the recent existing work (Zhang and Matsen 2019) is an adaptation of the NF to the phylogenetic tree problem. Motivated by the problem, each NF’s transformation is designed to be permutation equivariant. The authors adopt two ways of using normalizing flow (Planner and RealNVP) to this problem.

Strengths: 1-Calculation of approximation and amortization gaps in table 2 for different variational families 2-Developing permutation equivariant transformation for NF

Weaknesses: Main comments: The paper is difficult to follow because some details are just referred to previous work and are not mentioned in the paper. Please consider adding more details about the background, at least to the appendix. Since the main reason for using VI is to speed up the Bayesian inference, it is really important to provide experimental results regarding the time complexity of the method and its convergence. I think adding a plot similar to figure 4 in the paper (Zhang and Matsen 2019) can be helpful. Since your paper provides a more flexible family, it is expected not to converge as fast as (Zhang and Matsen 2019), but it is essential to see how much it works better than MCMC. Minor comments In the equation after line 248, the notation q borrowed directly from the paper Chris et al. (2018) to refer to variational distribution; however, it is better to use another notation to avoid confusion with branch length parameters. Also, the notation w_{x_i} in equation 5 is better to be changed to w_{n(x_i)} because ‘w’ does not depend on the value of x_i. It will be valuable to report the ELBO (k=1) for both PSP and the proposed method on the real datasets and showing that there is gain. After the author's response: I thank the authors for providing more empirical results for their model's computational complexity and flexibility. I think it would be better also to elaborate on the novelty of the method more.

Correctness: Yes

Clarity: More detail about the background is needed

Relation to Prior Work: Yes

Reproducibility: No

Additional Feedback:


Review 2

Summary and Contributions: The paper presents a new type of variational Bayesian phylogenetic Inference that makes use of normalizing flows to construct more expressive posteriors for branch length. A permutation equivariant construction of normalizing flows (planar and RealNVP) is proposed to handle the non-Euclidean branch length of phylogenetic models. Results on benchmark datasets demonstrate the effectiveness of the proposed family of posteriors.

Strengths: The paper presents a more expressive model for branch length approximation in phylogenetic inference. The proposed construction of permutation equivariant normalizing flows has grounds and connections to DeepSets, a SOTA theoretical framework for equivariance in deep models. The proposed variational inference approach is a faster and more efficient alternative to existing sampling-based methods with the additional advantage of providing a tighter bound. The proposed permutation equivariant construction of flows and variational inference of phylogenetic trees is relevant to the NeurIPS community as a faster and more efficient alternative to sampling-based methods.

Weaknesses: Lack of experimental evidence for significant improvements in lower bounds and marginal likelihood estimates compared with the existing diagonal Lognormal branch length approximation. The bijectivity of flow-based models mandates learning mappings in the same dimensionality of the posterior space. There is no discussion about such dimensionality and whether this introduces risks of overfitting. Experiments lack comparisons to sampling-based methods.

Correctness: Flow-based models are bounded by the manifold structure of the posterior distribution as it has issues modeling discontinuous manifolds, potentially assigns mass to unsupported regions. Given that, it is not clear if the flow model is the ideal candidate for the phylogenetic posteriors, but it improves the performance, though marginally.

Clarity: The paper is well written, structured, and easy to follow with enough background material.

Relation to Prior Work: The proposed equivariant constructions of flow models assume sets of scalars, which could be considered as a special case of sets of vectors. The connection to existing works that extend flow models to unordered data and whether they can handle sets of scalars are not well justified.

Reproducibility: Yes

Additional Feedback:


Review 3

Summary and Contributions: This paper proposes permutation equivariant normalizing flows as a variational inference method for phylogenetic tree estimation. While there is prior work that has done variational inference for phylogenetic tree estimation, the novelty comes from 1) using normalizing flows and 2) permutation equivariance in the architecture. [Comments in brackets denote post-rebuttal responses. My score remains unchanged at "Good submission. Accept," and I appreciate the authors' rebuttal as they clarified many of my questions.]

Strengths: Relevance: Variational inference and its applications is pretty core to the community to me. Significance: While not really a core ML subfield, phylogenetics and phylogenetic tree inference more specifically is really a fascinating and difficult problem with lots of real world importance. Theoretical grounding: This paper goes beyond a fair amount of ML applied to specific problem papers by taking the time to think about what sorts of inductive biases are needed for the problem at hand, and devises a permutation equivariant normalizing flow architecture for that. They also spend some time proving that the flows they use are equivariant. Empirical evaluation: This really isn't my subfield but the results look statistically significant; there are also some of the obvious ablation studies that I would have wanted (number of layers, knocking out the effect of the tree, etc.). The performance differences between flows and standard VI are clearly pointed out in Figure 3 which is really nice. Novelty: Beyond just solving a really nice applied problem, this is one of the first papers to really devise and study equivariant normalizing flows. Some googling points to https://proceedings.icml.cc/static/paper_files/icml/2020/6711-Paper.pdf as an extremely recent normalizing flows paper in this vein (not necessary to cite), but your work goes pretty far beyond that to encode a different type of symmetry and to use it for an inference problem.

Weaknesses: Relevance: The application is really quite niche which could mean that the equivariant architecture solution could get lost in the swamp of NeurIPS papers. Significance: I would really have hoped that the authors devoted a little bit more time to answering the following two claims: 1) Why is phylogenetic inference interesting to the broader ML community, and more specifically what can we do with better phylogenetic inference tools? Or, is there a larger problem that the development of this tool helps to solve? [Thanks for the clarification. I hope that some of the extra space for the camera ready goes to discuss your response.] 2) Why exactly is the "diagonal Lognormal branch design distribution ... not ... flexible enough"? My understanding is that MCMC methods are essentially the state of the art in this area, so is there evidence for substantial correlation between branches for these runs? [Thanks for the argument, but what I was really hoping for was some sort of traceplot demonstrating that the MCMC (and flow) methods actually pick up correlations over these parameters...] Empirical Evaluation: See point 2) above. You probably need to justify the usage of the importance sampling method in the VI approximation a bit better, although some googling on my end finds that the importance sampling from the VI estimate works pretty well - https://escholarship.org/content/qt77d8v106/qt77d8v106.pdf. The natural choice in the ML community is probably annealed importance sampling. [Thanks for the clarification and promise of experiment.] From a ML perspective, it's well known that normalizing flows produce really high quality likelihood estimates (Section 6.1 https://arxiv.org/pdf/1912.02762.pdf), which could explain why adding new layers just seems to continue decreasing the MLL. Is there a tradeoff between the number of layers and the MLL beyond which adding new layers to the flow just stops improving the likelihood (or it even becomes untrainable)? [Somewhat unaddressed.] Novelty: Although this is one of the first papers to study equivariant normalizing flows, it's not the first, and Section 5.6 of https://arxiv.org/pdf/1912.02762.pdf contains the broad portions of your proofs. However, they broadly just point to two workshop papers (one of which seems to have been extended into the ICML paper cited above).

Correctness: From reading the two proofs, things look broadly correct. I think that the empirical methodology is pretty much fine. I'd like to see a time comparison compared to standard VI if possible. I've trained some flows for inference previously and have been disappointed at how slow they are, which could prove problematic for the claims of "speed compared to MCMC." [Thanks for the iteration complexity comparison.]

Clarity: Yes, the paper is well-written and flows nicely throughout. I think the arrow in the amoritization gap pointer in Figure 3 could be pushed slightly higher. The figures are of good size and the captions tell the story of the figures well.

Relation to Prior Work: Yes, in general, it seems to be well related to the prior ML works in this area. I don't know the phylogenetic literature that well, but it seems to be novel in that space too. It seems like there's broadly some concurrent work on equivariant normalizing flows for physics problems that overlaps a bit (basically just Section 5.6 of https://arxiv.org/pdf/1912.02762.pdf that summarizes two NeurIPS '19 workshop papers). However, this isn't really a major deal at all.

Reproducibility: Yes

Additional Feedback: You probably need to reference the fact that the proofs to the propositions are in the Appendix right after the statements; it took me a few minutes to find them. Overall, I really like this work and think that it represents an improvement on the current state of the art. In the rebuttal, I'd like to see 1) a bit more discussion of how this work solves open phylogenetic problems and 2) a time comparison to VI and/or MCMC. Minor comments: Line 29 of appendix: "numerical stable" --> numerically stable. Line 30: is the determinant invariant for the same reason as the Jacobian being invariant?


Review 4

Summary and Contributions: In this work the authors consider the problem of creating normalizing flows as the approximating distribution for branch lengths in (Bayesian) phylogenetic variational inference problems. They begin referencing previous work that considers building approximating distributions based on tree splits and the associated primary subsplit pairs (PSPs). Under the existing framework, a log normal variational distribution is established, where the variational parameters for the mean, variance are composed by summing over variational parameters associated with splits and PSPs, thus making inference amortized. The overall trick for introducing flows into this problem seems twofold: (i) instead of establishing flows-specific variational parameters, use parameters associated with the splits and PSPs as per previous work as the parameters of the NF, and (ii) ensure the flows are permutation invariant since there is no consistent alignment for non-pendant tree edges. The authors introduce permutation-invariant versions of previously established flows (planar and RealNVP) with the parameters corresponding to split and PSP specific variables. They discuss inference methodology, and apply it to a number of real datasets for phylogenetic reconstruction problems, demonstrating state-of-the-art lower bounds estimates.

Strengths: Overall I think this is a very strong paper: it blends flow based/change of measure inference methods (an extremely active area of research) in a novel manner, applying it to a particular domain/inference problem (phylogenetics). The paper is methodologically exciting and achieves state of the art results, and acts as a strong foundation for any future work that would attempt to fit the entire inference procedure (topology + branch lengths) into a normalizing flow framework, rather than just the branch lengths.

Weaknesses: The only weakness I would leverage (and it’s very minor) is that I don’t think in any case is the marginal likelihood significantly improved over the previous (PSP) model (table 1) when the error bounds are taken into account (though admittedly this will be high variance), and in some cases it appears PSP achieves a higher marginal likelihood (on datasets DS5,6,8). Can the authors comment on this? Admittedly all variance estimates are lower.

Correctness: To the best of my knowledge the paper looks correct.

Clarity: Exceptionally minor comment: presumably \psi^\sigma is constrained > 0 ? (line 124) Equation 5: the authors introduce (z,x) presumably to standardize with existing NF notation but x is typically continuous leading to gamma and w being indexed by a continuous value, which is somewhat confusing, and doesn’t match eq 6 where x -> q, and the indexing is performed on edges e rather than on x. Can the authors clarify?

Relation to Prior Work: Yes — the paper well references and explains previous work on variational inference for phylogenetics. The paper appears to be the first of its kind connecting normalizing flows to Bayesian phylogenetic inference.

Reproducibility: Yes

Additional Feedback: Update: the authors addressed my (minor) concerns and I keep my score unchanged at 8

[Author Response · NeurIPS 2020]

We are very grateful to the reviewers for reading the manuscript in detail and providing helpful comments. Before
providing detailed response for each reviewer, we would like to address two common issues raised by the reviewers.

• We thank the reviewers for the suggestion of
reporting the experimental results regarding time
complexity of the method and its convergence. Re-
sults on DS1 are shown in the right figure. We see
that by the time PSP converges, the proposed flow-
based methods achieve comparable (if not better)
approximation quality and quickly surpass PSP as
the number of iterations increases. This means the
slower convergence of flow-based methods may
not affect the quality of approximation given sim-
ilar computation budget as that of PSP. We will
add direct comparison to MCMC in our revision.

* The computation budget ($10\times400000$) is the same as in Figure 4, Zhang and Matsen 2019 ($20\times200000$).

• Regarding marginal likelihood estimates, we want to clarify that all methods provide estimates for the same marginal
likelihood, and better approximation would lead to smaller variance. We will add comparisons to the stepping-stone
(SS) method (one of the state-of-the-art sampling-based methods for marginal likelihood estimation) in our revision.

Reviewer 1:

• We will add more details about the background to the appendix and clarify notation accordingly.

• The ELBO (K=1) for all methods are reported in the following table. The gain is more significant now.

| | Data set | DS1 | DS2 | DS3 | DS4 | DS5 | DS6 | DS7 | DS8 |
|---|---|---|---|---|---|---|---|---|---|
| LB (K=1) | PSP | -7111.23(1.04) | -26369.63(0.69) | -33736.60(0.33) | -13332.37(0.54) | -8218.35(0.20) | -6729.27(0.50) | -37335.15(0.11) | -8655.48(0.38) |
| | Planar(16) | -7110.33(0.16) | -26368.80(0.27) | -33736.14(0.14) | -13331.92(0.11) | -8217.98(0.13) | -6728.89(0.18) | -37334.78(0.11) | -8655.15(0.17) |
| | Planar(32) | -7110.22(0.17) | -26368.69(0.23) | -33736.02(0.21) | -13331.73(0.12) | -8217.90(0.14) | -6728.68(0.19) | -37334.60(0.12) | -8654.97(0.16) |
| | RealNVP(5) | -7110.12(0.13) | -26368.75(0.24) | -33735.86(0.10) | -13331.71(0.11) | -8217.80(0.14) | -6728.54(0.15) | -37334.44(0.11) | -8654.62(0.13) |
| | RealNVP(10) | **-7109.80(0.11)** | **-26368.59(0.23)** | **-33735.81(0.12)** | **-13331.39(0.08)** | **-8217.56(0.12)** | **-6728.04(0.14)** | **-37333.94(0.09)** | **-8654.02(0.12)** |

Reviewer 2:

• We want to clarify that the lower bounds are indeed significantly improved considering the marginal likelihoods they
approach. Moreover, the multi-sample lower bound (K=10) we reported is biased towards the marginal likelihood,
making improvements seem less significant. As suggested by Reviewer 1, we report the ELBO (K=1) in the table above,
which shows more significant gain of the proposed methods. Issues related to the marginal likelihood estimates are
discussed in the beginning. The structured parameterization of our flow-based models allows tree topologies to have
their own parameters for flows, while sharing similarities according to the local topological structures they possess.

• The existing flow models for unordered data require splitting the feature vector of each data point for RealNVP, which
does not work if these features are scalars (e.g., the branch lengths for edges of a phylogenetic tree). We will clarify this.

Reviewer 3:

• Phylogenetic analysis of viral genomes provides key insight into disease pathophysiology, spread and potential
control (e.g., the recent COVID-19 pandemic), offering invaluable information for public health decisions. Our methods
can perform Bayesian phylogenetic inference in a timely manner that is unlikely to be met by optimizing existing
MCMC strategies. Moreover, phylogenetic models are challenging for traditional methods and would inspire novel
statistical/machine learning approaches for models with complex, highly-structured, non-Euclidean parameter space.

• Yes, there are strong correlations among branch lengths. In fact, the significant lower bound improvement of RealNVP
(see Figure 3), which is based on coupling layers, is a strong evidence of the existence of such correlations.

• We thank the reviewer for introducing interesting recent work on equivariant normalizing flows. We will add them to
the related work section in our revision. We will add comparison to SS to better justify importance sampling using VI.

Reviewer 4:

• Issues regarding the marginal likelihood estimates are discussed in the beginning.

• Yes, $\psi^\sigma > 0$ and this constraint can be removed via the exponential transformation. $\gamma_{x_i}, w_{x_i}$ are parameters associated
with the input variable $x_i$, and they do not depend on the value of $x_i$ (echoed by Reviewer 1). For phylogenetic trees,
the branch lengths $\boldsymbol{q}$ are associated with the edges, and so are the parameters. We will clarify notation accordingly.

[Meta-Review · NeurIPS 2020]

This paper adapts normalizing flows with permutation invariance to phylogenetic trees. Reviewers were all happy with the submission and the author response, which combines an interesting application with natural methodological contributions. Congratulations on this nice work!